# Telemonitoring for perioperative care of outpatient bariatric surgery: Preference-based randomized clinical trial

E. S. van Ede[1,2]☯*, J. Scheerhoorn[3]☯, M. P. Buise[1], R. A. Bouwman[1,2], S. W. Nienhuijs[3]

**1** Department of Anesthesiology, Catharina Hospital Eindhoven, Eindhoven, The Netherlands, **2** Department of Electrical Engineering, Signal Processing Systems, Eindhoven University of Technology, Eindhoven, The Netherlands, **3** Department of Surgery, Catharina hospital Eindhoven, Eindhoven, The Netherlands

☯ These authors contributed equally to this work.
* lisa.v.ede@catharinaziekenhuis.nl

**Data Availability Statement:** The file ia available from the Open Science Framework database Accession number: DOI 10.17605/OSF.IO/256MN).

## Abstract

### Importance

Implementation of bariatric surgery on an outpatient basis is hampered by concerns about timely detection of postoperative complications. Telemonitoring could enhance detection and support transition to an outpatient recovery pathway.

### Objective

This study aimed to evaluate non-inferiority and feasibility of an outpatient recovery pathway after bariatric surgery, supported by remote monitoring compared to standard care.

### Design

Preference-based non-inferiority randomized trial.

### Setting

Center for obesity and metabolic surgery, Catharina hospital Eindhoven, the Netherlands.

### Participants

Adult patients scheduled for primary gastric bypass or sleeve gastrectomy.

### Interventions

*S*ame-day discharge with one week ongoing Remote Monitoring (RM) of vital parameters or Standard Care (SC) with discharge on postoperative day one.

### Main outcomes

Primary outcome was a thirty-day composite Textbook Outcome score encompassing mortality, mild and severe complications, readmission and prolonged length-of-stay. Non-inferiority of same-day discharge and remote monitoring was accepted below the selected

**Funding:** Philips Electronics Nederland BV, acting through research, Eindhoven, NL supplied the wearables used in this study, of which information is included in the protocol. Philips Eindhoven did not play a role in any of the following: The study design, data collection, analysis, decision to publish or preparation of the manuscript.

**Competing interests:** The authors have declared that no competing interests exist.

margin of 7% upper limit of confidence interval. Secondary outcomes included admission duration, post-discharge opioid use and patients' satisfaction.

## Results

Textbook Outcome was achieved in 94% (n = 102) in RM versus 98% (n = 100) in SC (RR 2.9; 95% CI, 0.60–14.23, p = 0.22). The non-inferiority margin was exceeded which is a statistically inconclusive result. Both Textbook Outcome measures were above Dutch average (5% RM and 9% SC). Same-day discharge reduced hospitalization days by 61% (p<0.001) and by 58% with re-admission days included (p<0.001). Post-discharge opioid use and satisfaction scores were equal (p = 0.82 and p = 0.86).

## Conclusion

In conclusion, outpatient bariatric surgery supported with telemonitoring is clinically comparable to standard overnight bariatrics in terms of textbook-outcome. Both approaches reached primary endpoint results above Dutch average. However, statistically the outpatient surgery protocol was neither inferior, nor non-inferior to the standard pathway. Additionally, offering same-day discharge reduces the total hospitalization days while maintaining patient satisfaction and safety.

## Introduction

Metabolic surgery accounts for a significant portion of healthdot resources for perioperative care. The high volumes of these procedures results in thousands of admissions per year [1], with the associated workload, patient burden [2] and hospital costs [3]. Enhanced Recovery After Bariatric Surgery protocols reduced the length of hospital stay after bariatric surgery without negative impact on morbidity or mortality [4]. The next step is a further reduction of length of stay to ambulatory [5] care. However, implementation of bariatric surgery on an outpatient [5] basis is hampered by concerns about timely detection of postoperative complications [6–8]. Severe events such as bleeding or anastomotic leakage requiring short term re-intervention occur in around 0.5–5% [9, 10] and are most often manifested by changes in vital signs, such as tachycardia and increased respiration rate [11, 12]. Therefore, patients frequently are hospitalized overnight for observation.

Non-invasive monitoring devices can be used remotely after bariatric surgery [13, 14] and could support the transition to an outpatient recovery pathway. It was recently suggested that same-day discharge after gastric bypass with additional remote monitoring is feasible [15]. However, this study was conducted on a relatively small and select population using non-continuous remote measurements operated by the patients themselves which could increase the risk of missing data which limits trend analysis. Additionally, recent evidence suggests the advantages of continuous measurement of vital signs over intermittent measurements [16–18]. Nevertheless, despite these early promising results, studies on the feasibility and safety of using remote monitoring devices for patients after surgery remain limited.

The aim of this patient preference randomized trial was to establish and evaluate non-inferiority and feasibility of an outpatient recovery pathway after bariatric surgery, supported by remote monitoring compared to standard care.

## Methods

### Trial design

This preference-based non-inferiority study was designed to compare the outcomes of two different recovery pathways after bariatric surgery in the Catharina Hospital, Eindhoven, The Netherlands. A detailed protocol of the study has been published [19]. The trial was registered at ClinicalTrials.gov (Identifier NCT04754893), and approved by the Medical Ethical Committee of the Maxima Medical Center, the Netherlands (Reference number W20.095). Standard care (SC) was discharge on postoperative day one after one overnight admission. In the intervention group, patients were discharged the same-day which was supported by remote monitoring (RM) using the Healthdot system (Philips Electronic Nederland BV) for seven days after surgery.

### Trial population

Patients were eligible to participate if scheduled for a primary Roux-en-Y gastric bypass or sleeve gastrectomy, if they were 18 years and older, had no allergy to white plasters, no pacemaker and someone nearby in their household the first night. After given written informed consent, patients were allocated to one of two trajectories conforming their preference. Patients without preference were randomly assigned to a study group. For this, the researchers used an online randomization software (random.org/lists/) that allocated patients based on the spots still available at that time.

### Telemonitoring

The Healthdot is a validated wearable data logger [13]. Attached mid-clavicular on the lowest left rib on the chest, it collects heart rate, respiratory rate, activity and posture continuously by means of accelerometer signals. Every five minutes, the mean of these values was transmitted and visualized in trend graphs in the Guardian Intellivue dashboard (Philips Electronic Nederland BV). A warning score was calculated for events with a heart rate and respiration rate above a pre-defined threshold of 110 beats per minute and 20 breaths per minute respectively [20]. These values were based on literature and clinical experience of the medical team. If the warning score remained elevated for 15 minutes, which means 3 data points, a notification was generated.

### Procedures

Patients from both study groups received equal perioperative care according to the Enhanced Recovery After Bariatric Surgery guidelines [21] and postoperative care in accordance with the hospitals' protocol for bariatric surgery. Patients allocated to the RM group were scheduled in the morning for the earliest time-slots. The Healthdot was applied in the recovery room directly after surgery. Discharge took place in the evening at the discretion of a nurse under supervision of a doctor. The patient had to feel well and be motivated to go home. Furthermore, hemoglobin decrease should not exceed 3.2 g/dL compared to preoperative value and the spot-check measured heart rate should be less than 100 beats per minute. Teleconsultation with a physician was scheduled on postoperative day one to assess physical condition. Vital signs were reviewed by the treatment-team every morning for up to seven days postoperatively and assessed over the past twenty-four hours to determine if additional action was indicated. The treatment-team had access to the vital signs, including any notification, only when logged into the dashboard. There were no active notifications sent to the clinicians. If a notification was seen after logging in, the patient received an additional teleconsultation. In case a patient

contacted the hospital, the vital signs dashboard was also reviewed. Patients receiving standard care remained in hospital overnight and discharged under the same conditions as the patients in the RM group. If these criteria were not met, patients stayed on the ward with ongoing spot check and lab monitoring if needed.

## Outcomes

Included in analysis were patient characteristics, comorbidities and surgery details according to Dutch Audit for Treatment of Obesity (DATO) definitions [22]. The primary outcome was a combined measurement of mortality, mild and severe complications (Clavien-Dindo 2 and higher), hospital readmission or prolonged length of stay (>2 hospitalization days after surgery), within 30 days after surgery; also known as Textbook Outcome [23]. Meeting this Textbook Outcome means that the patient has shown a perfect convalescence. If Textbook Outcome was not met, the most severe complication was assigned per patient. The total number of hospitalization days including readmission days, were calculated for each study group and specifically for the two types of surgery in the RM group. Per postoperative day, the number of patients who were discharged on that day was determined. The reason for failure of same-day discharge and teleconsultations on postoperative day one was inventoried. In addition, patient satisfaction and use of pain medication after discharge were assessed as secondary outcome. Patient satisfaction was assessed by a questionnaire which was completed by all patients on the seventh' day after surgery. In one of the questions, an overall score of 0–10 that reflects the general satisfaction of the perioperative pathway from surgery to one week after surgery was asked.

## Statistical analysis

Sample size calculation was performed by using uncorrected chi-square statistic and resulted in 97 patients per group to test with 80% power and an expected proportion of 0.96 whether Textbook Outcome of RM would not be inferior to SC [22]. In current practice, 95% of the patients who undergo bariatric surgery at the Catharina Hospital in Eindhoven meet this outcome measure. Related to the Dutch average of 88.7% [23], a non-inferiority margin of 7% was determined. The calculated non-inferiority margin for the relative risk (RR) difference from the assumed number of events was 2.75 (4%+7%)/4% ([expected event rate control group + non-inferiority margin] / expected event rate) [24]. All endpoint analyses were performed according to the intention-to-treat (ITT) principle [25]. In addition, a per protocol (PP) analysis was performed for the primary outcome. Results of the primary outcomes were presented in RR and in absolute risk difference (ARD) with a two-sided 95% confidence interval (CI) (Consort 2010, non-inferioty and equivalence trials, 17b). Non-inferiority was declared if the upper limit of the 95% CI of RR was < 2.75 and the ARD was < 7%. To demonstrate statistical significance of non-inferiority, p-value of <0.05 is needed (one-sided), calculated by using an asymptomatic score test (Wald's method) [26].

Interim analysis occurred after complete follow-up of 50 patients in each study group. Based on the results, it was decided by the research team whether the study could continue. Differences between group characteristic means, satisfaction scores were assessed with the use of an independent t-test and the total admission days with a non-parametric test (Mann-Whitney U). Differences between proportions of group characteristics were assessed with a chi-square or Fisher's exact test. Group means, admission duration, opioid use and satisfaction scores were considered statistically different at a two-tailed p-value less than 0.05. All analysis were performed by the research team with support of in-hospital statistician and clinical epidemiologist using IBM SPSS Statistics version 25.0 and the sample size calculation by Power and Sample Size Calculation software (v3.1.2, Vanderbilt University).

## Results

### Population

From March 2021 until November 2021, a total of 336 participants underwent screening of whom 208 consented participation and were asked for their preference. Of these patients, 103 preferred RM and 102 preferred SC. Additionally, three patients without a specific preference were randomly assigned into to the RM (n = 2) and SC group (n = 1). In total, 202 patients were analyzed (RM n = 102, and SC n = 100). Fig 1 shows an overview of the enrollment, treatment and follow-up. Baseline patient characteristics were comparable in both groups and are summarized in Table 1.

In total 102/102 patients were included for ITT and 66/102 for PP analyses as not all patients managed to be discharged the day of the surgery (see also Table 2 and hospitalization days paragraph).

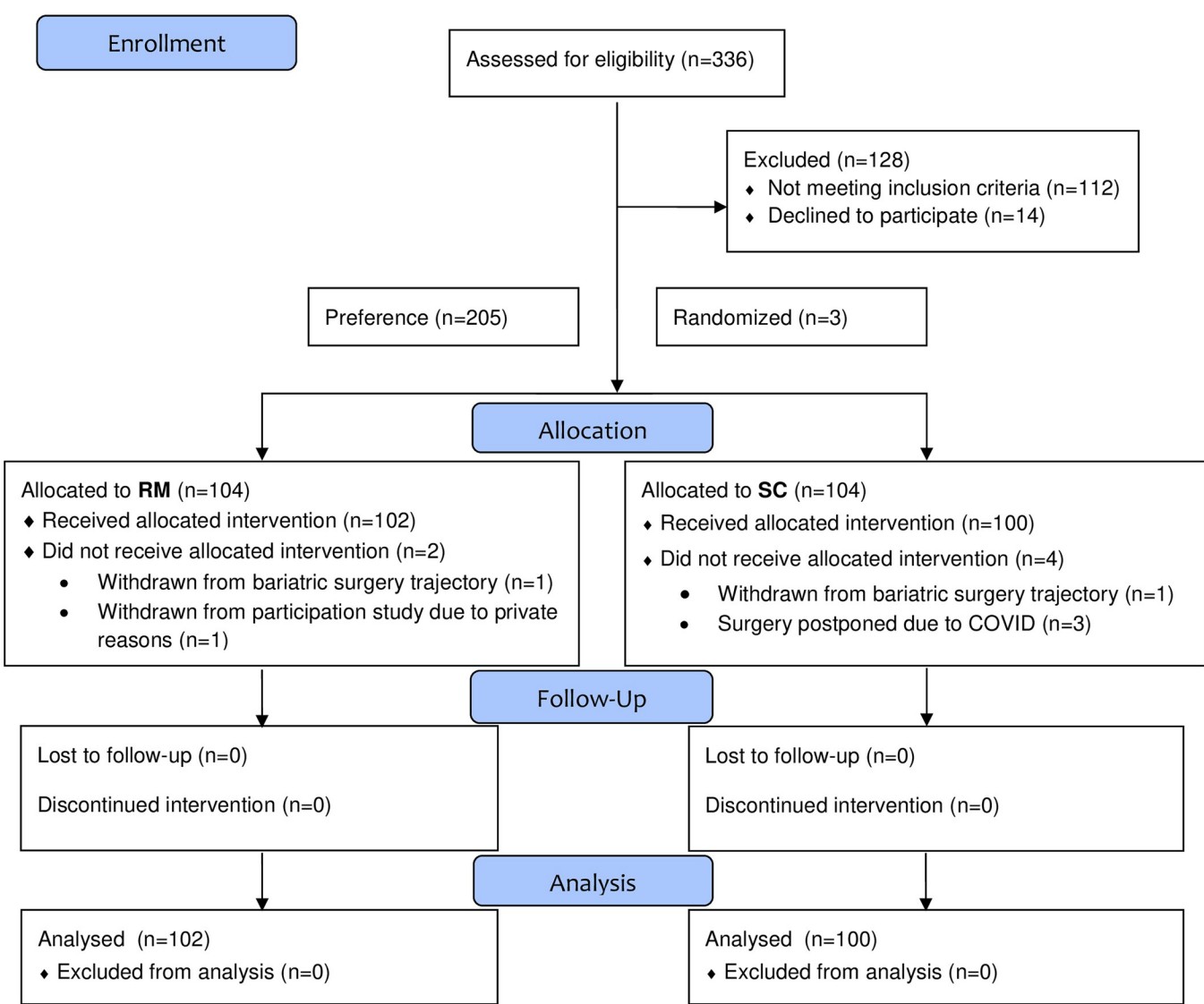

**Fig 1. Enrollment, randomization, and follow-up (Consort 2010 Flow Diagram).** RM = remote monitoring group. SC = standard care group.

**Table 1. Patient characteristics.**

| Patient Characteristics | | |
|---|---|---|
| Characteristic | **RM** | **SC** |
| | (n = 102) | (n = 100) |
| Age—years | 40 ± 11.3 | 41 ± 11.7 |
| Female–N (%) | 83 (81) | 86 (86) |
| Male–N (%) | 19 (19) | 14 (14) |
| Median body mass index § (IQR) | 42 (39–46) | 41 (39–44) |
| ASA–N (%) | | |
| 2 | 22 (22) | 20 (20) |
| 3 | 80 (78) | 79 (79) |
| 4 | 0 (0) | 1 (1) |
| Hypertension–N (%) | 25 (24.5) | 30 (30) |
| Diabetes Mellitus type II–N (%) | 7 (7) | 13 (13) |
| Gastroesophagal reflux disease–N (%) | 14 (14) | 16 (16) |
| Musculoskeletal pain–N (%) | 28 (28) | 35 (35) |
| Obstructive Sleep Apnea Syndrome–N (%) | 8 (8) | 14 (14) |
| Surgery type–N (%) | | |
| Sleeve gastrectomy | 54 (53) | 49 (49) |
| Roux-en Y Gastric Bypass | 48 (47) | 51 (51) |
| Surgery duration—minutes | 53 ± 17.5 | 52 ± 15.6 |

There were no statistical differences between the study groups. No data was missing. Plus-minus values are means ±SD. §Body mass index is the weight in kilograms divided by the square of the height in meters.

## Primary endpoints

Textbook Outcome was achieved in 96/102 (94%) in the RM group versus 98/100 (98%) in the SC group (Table 2). The PP result was 62/66 (94%) in the RM group versus 98/100 in the SC group. According to ITT analysis, the RR comparing Textbook Outcome measures between RM and SC was 2.9 (95% CI, 0.60–14.23, p = 0.22) and ARD 3.8% (95% CI, -1.45–9.2). For PP, a RR of 3.0 (95% CI, 0.57–16.07, p = 0.26) and ARD of 4.1% (-2.3–10.44) was calculated. The upper limits of the 95% CI crossed the specified non-inferiority margins, indicating that the results were statistically inconclusive [27]. Further, Textbook Outcomes of both groups were above the Dutch average of 88.7% (RM 5.3% and SC 9.3%).

**Table 2. Primary endpoint (textbook outcome) according to ITT analysis.**

| | **RM** | **SC** | **Risk ratio (95% CI)** |
|---|---|---|---|
| | **n = 102** | **n = 100** | **P-value** |
| **Textbook Outcome** | **94%** | **98%** | 2.9 (0.60–14.23) p = 0.22 |
| | N (%) | N (%) | |
| Mortality | 0 | 0 | |
| Severe complications | 3 (3) | 0 | |
| Mild complications | 2 (2) | 1 (1) | |
| Readmission | 1 (1) | 1 (1) | |
| Prolonged length of stay | 0 | 0 | |

No data was missing.

**Table 3. Hospitalization days.**

| | RM | | SC | | P-value |
|---|---|---|---|---|---|
| | Patients | Days | Patients | Days | |
| 1 day | 30 | 30 | 88 | 88 | |
| 2 days | 5 | 10 | 12 | 24 | |
| 4 days | 1 | 4 | 0 | 0 | |
| Readmission days | 3 | 6 | 1 | 2 | |
| Total days | | **50** | | **114** | P < 0.001 |

The total number of admission and readmission days and the difference between study groups is presented. 'Patients': number of patients discharged on the corresponding postoperative day. 'Days': cumulative number of days spend in the hospital by these patients.

No mortality was encountered. Mild complications registered in the RM group were a prescription of oral antibiotics to treat an infected hematoma and a endoscopic clipping on postoperative day one requiring readmission after initial same-day discharge. In the SC group, a Computed Tomography of the cerebrum was performed on a patient who had fallen on postoperative day one, while still admitted in hospital. The scan showed no abnormalities. Severe complications included three patients from the RM group who required re-surgery due to postoperative bleeding. Two of these patients were readmitted on postoperative day two after initial same-day discharge. The third patient underwent re-surgery on postoperative day two and remained in hospital for four consecutive days. These patients with a complicated postoperative course did not use anticoagulation medication. Beside these complications, none of the patients had a prolonged hospital stay. In the RM and SC groups each, one patient was readmitted on postoperative day one after teleconsultation and on day four, respectively. Both readmissions were for observation of nausea and vomiting (Clavien-Dindo 1). There was no need to administer intravenous fluids.

## Hospitalization days

Discharge on postoperative day zero was achieved in 66/102 (65%) of RM patients. In 30/102 (29%), the patients went home on postoperative day one. The total amount of hospitalization days was significantly lower in the RM group (61% p<0.001 concerning initial hospital stay and 58% p<0.001 when including readmissions) (Table 3). No differences were found in the total number of hospital days of patients in the RM group who underwent a RYGB or an SG (21 versus 23 days, respectively).

## Patient satisfaction

The reasons not being discharged on the same-day for the 36 patients of the RM group are shown in Table 4. Patient's doubts and nausea were most common. The remaining patients (n = 66) received a scheduled teleconsultation postoperatively. Of this group, 12 patients had a total of 14 issues to report during that consultation. During the entire first postoperative day, a total of 22/66 patients (33%) had 26 issues to report that, in addition to the scheduled consultations, could be managed with 15 additional phone calls. Most commonly mentioned issues were pain and nausea. Except for two readmissions and one assessment in the clinic, all patients could be counseled or treated on a remote basis. Patients who received remote care were equally satisfied compared to patients in the SC group (8.0 ± 1.6 versus 8.0 ± 1.4 respectively p = 0.86, 95% CI -0.4–0.5). Satisfaction scores were missing of 12 patients in SC group and 3 patients of the RM group who were discharged the same-day. In addition, patients from

**Table 4. Reasons for not achieving same-day discharge versus issues during teleconsultation.**

| RM–no. (%) | |
|---|---|
| **Reasons for not achieving same-day discharge** | |
| • Tachycardia | 5 (5.0%) |
| • O2 suppletion | 2 (2.0%) |
| • Surgical | 1 (1.0%) |
| • Nausea | 9 (8.8%) |
| • Pain | 5 (4.9%) |
| • Not comfortable with discharge | 12 (11.8%) |
| • No transport available (1) | 1 (1.0%) |
| • Diabetes protocol not stopped in time (1) | 1 (1.0%) |

| **Issues during teleconsultations on postoperative day 1 (66 patients)** | |
|---|---|
| Issue | Care |
| Nausea 8 (12%) | • Conservative care 1 (12.5%)<br>• Anti-emetics 5 (62.5%)<br>• Follow-up* 1 (12.5%)<br>• Readmission 1 (12.5%) |
| Pain 14 (21%) | • Conservative care 6 (43%)<br>• Follow-up* 2 (14%)<br>• Opiods 5 (36%)<br>• Assessment clinic 1 (7%) |
| Heartburn 1 (1.5%) | Conservative care 1 (100%) |
| Collapse 1 (1.5%) | Readmission 1 (100%) |
| Notification 2 (3%) | No action needed 2 (100%) |
| *Follow-ups 3 (4.5%) | Conservative care |

Upper: reasons failure same-day discharge of patients in the RM group, in % patients of the total RM group (n = 102) and number of patients (no.). Lower: issues during planned teleconsultations on postoperative day one, in % of total patients receiving teleconsultations (n = 66) and number of patients (no.). Follow-up: additional call to assess the result of advice or treatment given or to reassure the patient.

both groups were prescribed the same amount of opioids after being discharged (10/102 (11%) versus 9/100 (9%) in SC group p = 0.82).

## Adverse events

There were no adverse events during the study. Three of the 102 patients wearing Healthdot developed a minor rash on monitoring day 5 (n = 2) and day 7. After a physician's assessment of safety, the patch was removed from these patients.

## Discussion

In adult patients who undergo primary bariatric surgery, an outpatient pathway supported with telemonitoring is clinically comparable to standard overnight bariatrics in terms of text-book-outcome. Both approaches reached primary endpoint results above Dutch average. Also, the patient-preferred outpatient recovery pathway with remote monitoring resulted in significantly fewer hospitalization days without compromising patient satisfaction and safety.

Statistically the outpatient surgery protocol was neither inferior, nor non-inferior to the standard pathway. An explanation could be that the standard group had a higher outcome score (98%) than the expected 96% on which the sample size is based [24]. With the outcomes observed in the control group, a larger number of patients would have been required for ITT and especially PP analysis to establish non-inferiority. The number of events in both groups is

also very low, so that a small increase in the difference of Textbook Outcomes between groups has major consequences for the confidence interval.

This study evaluated same-day discharge for both sleeve gastrectomy and gastric bypass without any comorbidity restriction. Previous studies had a retrospective design, except for a recently conducted randomized controlled study [28], and focus on one specific technique, more patients' selection criteria and no comparison with an overnight hospital stay [6–8, 15, 28–32]. The outcome parameters used in previous research are comparable to the combined Textbook results in our current study.

No mortality was encountered, although the group size was small in this matter. Nevertheless, in review the mortality rates of outpatient post-bariatric patients did not differ statistically from patients hospitalized overnight, with the exception of Inaba and Morton et al [6–8]. The latter ones attributed the increase mortality to an increased number of cardiopulmonary complications requiring resuscitation, unplanned intubation and Intensive Care admission requirements. The authors hypothesized that this may be due to a poor assessment of the patient's risk prior to surgery and failure to rescue the patient in an unsupervised environment [6–8]. Extended monitoring of patients, during the most risky hours after surgery, could help clinicians detect complications. With higher data density, it may also even be possible to determine more accurately the timing of the discharge. However, the added value of continuous remote monitoring in post-bariatric patients is currently anecdotal as no data or literature is available yet regarding improving survival and morbidity.

No significant differences were found for readmission and reoperation. One patient was readmitted within the same-day discharge period because of collapse and abnormal trend analyses found at teleconsultation on postoperative day one. Two patients undergoing outpatient care required readmission on postoperative day two due to postoperative bleeding. As no abnormalities were revealed from teleconsultation and vitals on postoperative day one, they did not seem to be related to the same-day discharge pathway. Even an hypothesis of more straining at home was considered unlikely as a tachycardia was expected to occur within 8–24 hours of surgery [11] for these late bleedings. In both cases, treatment was not started too late. If standard care had been provided, the clinical course would not have been different since the patients could be discharged home anyway.

The current study showed a halved length of stay in the RM group, not excluding any patient groups and taking patient preference into account. Recently, Nijland et al. were the first to conduct a prospective study combining same-day discharge with remote monitoring. A success rate of 88 percent was achieved, which was higher than in our cohort. The authors noted that specific preparation in the preoperative pathway was a positive factor contributing to this success [15]. Comparable to the study of Aftab et al [5], our results showed quite some impact of patients hesitation regarding early discharge due to nausea. On the other hand patients with the same complaints however willing to go home achieved same satisfaction, suggesting that this is feasible. In our design there was less focus on preparation for same day discharge allowing an unstrained preference of the patients. Success rate is expected to increase if the expectation management of patients takes on a more central role in the implementation of the outpatient recovery trajectory and the remote monitoring device applied only after fulfilling the criteria to be discharged.

## Strengths

Clinical trials preferably include random assignment. However, implementation of results of such trials can be hampered by reduced external validity. Adding patient preferences can limit this effect without compromising internal validity [33]. Randomization of patients can actually

lead to bias, as there is a chance that the patient would not be motivated if the recovery trajectory was not preferred. In addition, besides including patients' preference mimics current daily practice, stimulating patient autonomy is important to empower confidence and self-care which may benefit short-term and long-term outcomes.

## Limitations

No real-time notifications were used during the telemonitoring period in the outpatient recovery pathway. Hypothetically, incorporating real-time alarms into daily practice may improve patient safety and outcome measures, while also negatively impacting workload and induce alarm burden. Future reports will evaluate the performance of the currently used follow-up and notification protocol on complication detection. The results of this study were applicable to Dutch bariatric surgical patients. It can be imagined that this may not be the case for every bariatric surgical center as the composition of patient characteristics varies.

The added value of continuous remote monitoring on morbidity and mortality after general or bariatric surgery has not been established yet. Large datasets and future studies are needed to elaborate on this aspect, but also to improve monitoring and alarming techniques [34–37]. Nevertheless, we envision that the results and experiences gained from the outpatient monitoring pathway in the present study, can serve as a blueprint for rolling out telemonitoring for other perioperative care pathways.

In conclusion, without any restriction on comorbidities, outpatient bariatric surgery supported with telemonitoring is clinically comparable to standard overnight bariatrics in terms of textbook-outcome. Both approaches reached primary endpoint results above Dutch average. However, statistically the outpatient surgery protocol was neither inferior, nor non-inferior to the standard pathway. Additionally, offering same-day discharge reduces the total hospitalization days while maintaining patient satisfaction and safety.

## Supporting information

**S1 Checklist.**
(PDF)

**S1 File. Study protocol.**
(PDF)

**S2 File.**
(PDF)

## Acknowledgments

This study was initiated from the Eindhoven MedTech Innovation Center (e/MTIC), the project is a collaboration of the Catharina Hospital in Eindhoven, Technical University of Eindhoven, and Philips Research of the Netherlands. The authors wish to thank the Philips research-team for their constructive and practical contributions and Philips Healthdot-team for technical support during clinical use of Healthdot.

The authors would like to express their gratitude to:

*Yentl Lodewijks and Friso Schonck*, Catharina Hospital Eindhoven, Eindhoven, The Netherlands: MD, PhD-candidates who helped conducting the study.

*Department of the Obesity Clinic and Surgery Planning*, Catharina Hospital Eindhoven, Eindhoven, The Netherlands: Supported planning concerning inclusion and surgery.

*Department of clinical physics and technical engineering*, Catharina Hospital Eindhoven, Eindhoven, The Netherlands: enabling implementation of new medical device and provide technical logistics support.

*Department of Pharmacy*, Catharina Hospital Eindhoven, Eindhoven, The Netherlands: enabling new pathway of dispensing postoperative medication.

*Saskia Houterman and Sylvie Kolfschoten* Department of education and research, Catharina Hospital Eindhoven, Eindhoven, The Netherlands: helped performing statistics.

*Marcel van 't Veer*; Research and development, hartcentrum & Wetenschapsbureau Catharina Hospital Eindhoven, Eindhoven, The Netherlands: helped performing statistics.

## Author Contributions

**Conceptualization:** E. S. van Ede, J. Scheerhoorn, R. A. Bouwman, S. W. Nienhuijs.

**Formal analysis:** E. S. van Ede.

**Investigation:** E. S. van Ede, J. Scheerhoorn.

**Supervision:** M. P. Buise, R. A. Bouwman, S. W. Nienhuijs.

**Writing – original draft:** E. S. van Ede, J. Scheerhoorn.

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
