## [Decision Letter · Decision Letter 0]

17 Nov 2022

PONE-D-22-26848Telemonitoring for perioperative care of outpatient bariatric surgery: preference-based Randomized Clinical TrialPLOS ONE

Dear Dr. van Ede,

Thank you for submitting your manuscript to PLOS ONE. After careful consideration, we feel that it has merit but does not fully meet PLOS ONE’s publication criteria as it currently stands. Therefore, we invite you to submit a revised version of the manuscript that addresses the points raised during the review process. Please submit your revised manuscript by Jan 01 2023 11:59PM. If you will need more time than this to complete your revisions, please reply to this message or contact the journal office at plosone@plos.org. Please include the following items when submitting your revised manuscript:A rebuttal letter that responds to each point raised by the academic editor and reviewer(s). You should upload this letter as a separate file labeled 'Response to Reviewers'.A marked-up copy of your manuscript that highlights changes made to the original version. You should upload this as a separate file labeled 'Revised Manuscript with Track Changes'.An unmarked version of your revised paper without tracked changes. You should upload this as a separate file labeled 'Manuscript'.

We look forward to receiving your revised manuscript.

Kind regards,

Steven Eric Wolf, MD

Academic Editor

PLOS ONE

Journal Requirements:

3. We note that the original protocol that you have uploaded as a Supporting Information file contains an institutional logo. As this logo is likely copyrighted, we ask that you please remove it from this file and upload an updated version upon resubmission.

Additional Editor Comments:

Editor - Thank you for submitting your paper to us for review. I sent it to seven distinguished referees for comment and decision of whom two agreed to review; you will see these below. They thought that the paper has merit, but each have raised some substantial issues to be addressed in a revision. Please carefully consider the comments below and reply directly to each in a cover letter with appropriate marked and linked changes to the manuscript. I look forward to seeing the revision, which I will send back to the same referees for further comment and decision. Please understand that this is not a guarantee of future publication, as the revised manuscript itself must stand on its own merit.

Reviewers' comments:

Reviewer's Responses to Questions

**Comments to the Author**

1. Is the manuscript technically sound, and do the data support the conclusions?

Reviewer #1: Yes

Reviewer #2: Yes

2. Has the statistical analysis been performed appropriately and rigorously? 

Reviewer #1: Yes

Reviewer #2: No

3. Have the authors made all data underlying the findings in their manuscript fully available?

Reviewer #1: Yes

Reviewer #2: No

4. Is the manuscript presented in an intelligible fashion and written in standard English?

Reviewer #1: Yes

Reviewer #2: Yes

5. Review Comments to the Author

Reviewer #1: I want to commend the authors on a well written manuscript. This article excels both from a topic addressed and content standpoint. We are in an era of increasing demand on the healthcare system. It is more important to than ever before to manage our resources while keeping excellent quality outcomes.

Please edit the first line of the introduction paragraph for it to make better grammatical sense. Other than that no issues or concerns identified.

Reviewer #2: Line 130-133 The description is not clear here. Those events mean “a perfect convalescent”?

Line 144 what test was used for the sample size calculation?

Line 153: Which non-parametric test was used? Need to clarify which outcome was compared using t-test and which was compared using the non-parametric test.

Line 155: It is not appropriate to use two -sided p value for the non-inferiority trial.

Line 152 & 156 what group means were mentioned here?

Line 178 the result of primary outcome comparison is not clearly written for a non-inferiority test. Line 181. What are the 95% confidence intervals for event rates? One sided or two-sided. The confidence interval of event rate should be reported in the text and in table 2.

Line 179 and table 2: What is the purpose to use risk ratio? The estimated risk ratio does not seem correct. Did you use odds ratio?

Line 180 & Table 2: what are the sample sizes for ITT and per-protocol? Is there any imputation used? If no data is missing (as specified in the title), what is the difference between the two?

Table 2. Add total N of each group. Omit risk ratio and add 95% CI for event rate. For mortality and etc., report N (%).

Line 224 all patients were prescribed. Why n=10 and 9 in RM and SC respectively?

Table 3 is confusing.

Table 4 add (%)

6. PLOS authors have the option to publish the peer review history of their article (what does this mean?). If published, this will include your full peer review and any attached files.

Reviewer #1: No

Reviewer #2: No

---

## [Author Response · Author response to Decision Letter 0]

12 Dec 2022

Dear Editors

Thank you for the opportunity to resubmit our manuscript. The comments of the reviewers were useful and have been addressed to accordingly, see below. Looking forward to hear from you.

Sincerely yours,

Lisa van Ede

On behalf of my co-authors.

Responses to the reviewers;

Reviewer #1

Please edit the first line of the introduction paragraph for it to make better grammatical sense. Other than that no issues or concerns identified. Thank you for this comment, we have altered this sentence. 

Reviewer #2:

Line 130-133 The description is not clear here. Those events mean “a perfect convalescent”? Indeed, a perfect convalescent, outcome as hoped and described in a textbook, we clarified this as a composite measure of clinical process indicators. Textbook Outcome is realized for those who reached all desired short-term health indicators. 

Line 144 what test was used for the sample size calculation? Sample size was calculated using an uncorrected chi-square statistic. This is added in the manuscript. 

Line 153: Which non-parametric test was used? Need to clarify which outcome was compared using t-test and which was compared using the non-parametric test. The non-parametric test that was used was a Mann-Whitey U test. This is added to the manuscript. Also, we clarified which test was used for what outcome. 

Line 155: It is not appropriate to use two -sided p value for the non-inferiority trial. Thank you for this comment, it is correct indeed that the results should be presented as one-sided p-values. The p-values in the first manuscript were results of chi-square tests. Now, these p-values were changed for p-value for non-inferiority testing using the Wald Method which is more correct. The numbers are altered in the abstract, main text and table 2. 

Wellek S. Statistical methods for the analysis of two-arm non-inferiority trials with binary outcomes. Biom J. 2005 Feb;47(1):48-61; discussion 99-107.

Line 152 & 156 what group means were mentioned here? Specifications of group means of patients characteristics have been clarified. 

Line 178 the result of primary outcome comparison is not clearly written for a non-inferiority test. The results in the paragraph for primary outcomes are rewritten using examples of existing literature and source 2

2. Piaggio G, Elbourne DR, Pocock SJ, Evans SJ, Altman DG; CONSORT Group. Reporting of noninferiority and equivalence randomized trials: extension of the CONSORT 2010 statement. 

JAMA. 2012 Dec 26;308(24):2594-604. 

Line 181. What are the 95% confidence intervals for event rates? One sided or two-sided. The confidence interval of event rate should be reported in the text and in table 2. The 95% CI was only calculated on the combined outcome measure because the individual event rates were too low. The result of the individual events would not be reliable as the primary outcome was a combined measure, in accordance with advice of consulted hospital's methodologist. For this reason, also the p-values behind the event rates were removed from table 2. The 95% CI of the combined textbook outcome is presented 2-sided. This information is added to the methodology section. 

Line 179 and table 2: What is the purpose to use risk ratio? The risk ratio was assessed and presented conform the consort guidelines for non-inferiority trials with binary outcomes (see link below). According to the guidelines, also the risk difference is added to the method and result section. 

http://www.consort-statement.org/checklists/view/650-non-inferiority-and-equivalence-trials/827-binary-outcomes

The estimated risk ratio does not seem correct. Did you use odds ratio? Correct, presented RR had to be adjusted. As a result also the Confidence Interval enlarged for which discussion and conclusion had to be adjusted. 

Line 180 & Table 2: what are the sample sizes for ITT and per-protocol? Is there any imputation used? If no data is missing (as specified in the title), what is the difference between the two? It is correct that no data was missing, this was not a distinctive in analysis. Per-protocol analysis focused on those who were actually discharged the same day. In the ITT approach also those who did not succeed to leave the same day were included. Additional info has been added to the method section. 

Table 2. Add total N of each group. Omit risk ratio and add 95% CI for event rate. For mortality and etc., report N (%). The table is adjusted except for 95% CI for event rates. The 95% CI was only calculated on the combined outcome measure because the individual event rates were too low. The result of the individual events would not be reliable as the primary outcome was a combined measure, in accordance with advice of consulted hospital's methodologist. Also for this reason the p-values behind event rates are removed. 

Line 224 all patients were prescribed. Why n=10 and 9 in RM and SC respectively? Correct, it was a typo error, altered into ‘In addition, patients from both groups were prescribed the same amount of opioids after being discharged (10/102 (11%) versus 9/100 (9%) in SC group p=0.82).’ 

Table 3 is confusing. The layout is upgraded for more clarification.

Table 4 add (%) Adjusted

---

## [Decision Letter · Decision Letter 1]

7 Feb 2023

Telemonitoring for perioperative care of outpatient bariatric surgery: preference-based Randomized Clinical Trial

PONE-D-22-26848R1

Dear Dr. van Ede,

We’re pleased to inform you that your manuscript has been judged scientifically suitable for publication and will be formally accepted for publication once it meets all outstanding technical requirements.

Kind regards,

Steven E. Wolf, MD

Academic Editor

PLOS ONE

Additional Editor Comments (optional):

Reviewers' comments:

Reviewer's Responses to Questions

**Comments to the Author**

1. If the authors have adequately addressed your comments raised in a previous round of review and you feel that this manuscript is now acceptable for publication, you may indicate that here to bypass the “Comments to the Author” section, enter your conflict of interest statement in the “Confidential to Editor” section, and submit your "Accept" recommendation.

Reviewer #2: All comments have been addressed

2. Is the manuscript technically sound, and do the data support the conclusions?

Reviewer #2: (No Response)

3. Has the statistical analysis been performed appropriately and rigorously? 

Reviewer #2: (No Response)

4. Have the authors made all data underlying the findings in their manuscript fully available?

Reviewer #2: (No Response)

5. Is the manuscript presented in an intelligible fashion and written in standard English?

Reviewer #2: (No Response)

6. Review Comments to the Author

Reviewer #2: (No Response)

7. PLOS authors have the option to publish the peer review history of their article (what does this mean?). If published, this will include your full peer review and any attached files.

Reviewer #2: No

---

## [Editor Report · Acceptance letter]

10 Feb 2023

PONE-D-22-26848R1 

Telemonitoring for perioperative care of outpatient bariatric surgery: preference-based Randomized Clinical Trial 

Dear Dr. van Ede:

I'm pleased to inform you that your manuscript has been deemed suitable for publication in PLOS ONE. Congratulations! Your manuscript is now with our production department. 

Kind regards, 

on behalf of

Dr. Steven E. Wolf 

Academic Editor

PLOS ONE